# Emerging Concepts in Defective Macrophage Phagocytosis in Cystic Fibrosis

**DOI:** 10.3390/ijms23147750

**Published:** 2022-07-13

**Authors:** Devi Jaganathan, Emanuela M. Bruscia, Benjamin T. Kopp

**Affiliations:** 1Center for Microbial Pathogenesis, The Abigail Wexner Research Institute at Nationwide Children’s Hospital, Columbus, OH 43205, USA; devi.jaganathan@nationwidechildrens.org; 2Department of Pediatrics, Yale University School of Medicine, New Haven, CT 06510, USA; emanuela.bruscia@yale.edu; 3Division of Pulmonary Medicine, Nationwide Children’s Hospital, 700 Children’s Drive, Columbus, OH 43205, USA

**Keywords:** cystic fibrosis, immune response, macrophage, microorganisms

## Abstract

Cystic fibrosis (CF) is caused by mutations of the cystic fibrosis transmembrane conductance regulator (CFTR) gene. Chronic inflammation and decline in lung function are major reasons for morbidity in CF. Mutant CFTR expressed in phagocytic cells such as macrophages contributes to persistent infection, inflammation, and lung disease in CF. Macrophages play a central role in innate immunity by eliminating pathogenic microbes by a process called phagocytosis. Phagocytosis is required for tissue homeostasis, balancing inflammation, and crosstalk with the adaptive immune system for antigen presentation. This review focused on (1) current understandings of the signaling underlying phagocytic mechanisms; (2) existing evidence for phagocytic dysregulation in CF; and (3) the emerging role of CFTR modulators in influencing CF phagocytic function. Alterations in CF macrophages from receptor initiation to phagosome formation are linked to disease progression in CF. A deeper understanding of macrophages in the context of CFTR and phagocytosis proteins at each step of phagosome formation might contribute to the new therapeutic development of dysregulated innate immunity in CF. Therefore, the review also indicates future areas of research in the context of CFTR and macrophages.

## 1. Introduction

Cystic fibrosis (CF) is the most common life-limiting autosomal recessive disorder of persons of Caucasian descent; however, the disease has recently been increasingly reported in other racial and ethnic groups where it is likely underdiagnosed [1]. CF is a multiorgan disease involving all organ systems, most notably the respiratory and gastrointestinal tracts [2]. Mutations in both alleles of the gene that encodes for the CF transmembrane conductance regulator protein (CFTR) on chromosome 7 are diagnostic for the disorder. Over 2000 CFTR variants exist and are grouped into six to seven classes [3]. The loss of CFTR expression in epithelial cells leads to a dehydrated airway surface liquid interface, resulting in a thick mucus layer and pathogen retention from poor mucociliary clearance. Besides the role of CFTR in defective ion transport and subsequent mucus formation [4], the expression of CFTR in immune cells and its regulation of various immune functions are increasingly reported [5]. Subsequently, CF is characterized by exaggerated inflammation that contributes to the progressive loss of lung structure and function [6].

Given the diverse role of macrophages in the inflammatory response including initiation, resolution, and repair, these cells are critical contributors to CF pathogenesis. There are many types of macrophages, both tissue-resident and recruited. Alveolar macrophages are sentinels of the airway host defense and must be able to ignore innocuous antigens, while simultaneously patrolling the airways to clear pathogens [7,8]. When a pathogen is encountered, alveolar macrophages mount an immune response to release cytokines and chemokines and recruit neutrophils and monocyte-derived macrophages (MDMs) that drive inflammation [9]. Macrophages are remarkably plastic immune cells; depending on the tissue environment and surrounding stimuli, they possess a variety of receptors and corresponding phenotypic features with different capacities for inflammatory, anti-inflammatory, and reparative functions [10]. Macrophages are early responders to invading pathogens and importantly, have been shown to contribute to defective bacterial clearance in CF [11]. Impaired pathogen internalization and intracellular killing by CF macrophages are commonly reported [12]. Phagocytosis is a complex process that requires numerous signaling pathways and structural rearrangements to occur. Effective phagocytosis depends on particle recognition by cell surface receptors, their internalization, and the subsequent maturation of phagosomes [13]. This review focused on (1) current understandings of the signaling underlying phagocytic mechanisms; (2) existing evidence for phagocytic dysregulation in CF; and (3) the emerging role of CFTR modulators in influencing CF phagocytic function.

## 2. Receptor-Initiated Phagocytosis

### 2.1. Pattern Recognition Receptors

The first step in phagocytosis involves the recognition of the particle by the phagocytic cell. Foreign particles such as microbial pathogens are recognized by structures conserved among microbial species termed pathogen-associated molecular patterns (PAMPs). Pattern recognition receptors (PRRs) sense microbes through PAMPs. Some receptors bind to pattern recognition molecules and prime the cells for phagocytosis but do not initiate phagocytosis. PRRs such as Toll-like receptors (TLR) prime the cells for phagocytosis, while PRRs such as Dectin-1, Mannose receptor (CD206), CD14, Scavenger receptor A (CD204), CD36, and MARCO induce phagocytosis [14,15,16,17,18,19]. However, some studies indicated that TLRs also induce phagocytosis [20].

### 2.2. Opsonic Receptors

Another mechanism by which phagocytes recognize foreign particles involves opsonins. Opsonins are soluble particles coated on microorganisms and are recognized by specific receptors on the phagocyte membrane. IgG and complement components are important opsonins that induce efficient phagocytosis. Several opsonic receptors that induce phagocytosis include Fcγ receptors (FcγRI, FcγRIIa, FcγRIIIa), FcαRI, FcεRI, CR1, CR3, CR4, and α5β1 receptors [19,21,22,23,24].

### 2.3. Apoptotic Receptors

Millions of dead cells are generated each day and need to be cleared by patrolling cells such as macrophages. Apoptotic bodies release substances not found in normal cells to assist with phagocyte recognition. The molecules expressed on the surface of apoptotic cells include ATP, lysophosphatidylcholine, sphingosine-1-phosphate, and phosphatidylserine [25]. Some of the apoptotic receptors that induce phagocytosis include T cell immunoglobulin -1 (TIM-1), TIM-2, CD36, ανβ3, ανβ5, and Stabilin-2 [26,27,28,29,30,31]. A detailed examination of apoptosis mechanisms is provided elsewhere [32,33,34].

### 2.4. Factors That Influence Phagocytosis

Phagocytosis is efficient when multiple receptors and ligands co-operate simultaneously. Phagocytosis is also influenced by the density and affinity of receptors and the relative mobility of receptors [35]. In CR-3-mediated phagocytosis, additional stimuli such as TLR, FcγR, and CD44 increase receptor affinity to their ligand [36,37,38]. There are a diversity of receptors and ligands that recognize target particles. FcγR and CR-3 are the major receptors extensively studied in macrophages.

### 2.5. Fcγ Receptors

FcγR are glycoproteins expressed on the phagocytic cell membrane that bind the Fc portion of various immunoglobulins with different affinities [39]. The coordination of FcγR signaling is important to regulate the cellular commitment to phagocytosis [40]. The activation of FcγR results in the phosphorylation of tyrosine-based activation motif (ITAM). Src-Family kinases mediate the phosphorylation of ITAM, creating a docking site for the tyrosine kinase Syk, which itself phosphorylates near ITAMs [41]. Syk further activates several downstream signaling pathways that trigger phospholipase C to produce second messengers IP3 and DAG. These second messengers induce calcium release to activate protein kinase C and the activation of extracellular signal-related kinases (ERK and p38) [42]. Syk also activates other pathways involving PI3K and Vav that regulate the next steps in phagocytosis such as actin polymerization, and pseudopod formation.

The mechanism of complement-mediated phagocytosis is not entirely understood; however, the role of Syk in mediating complement-mediated phagocytosis was previously demonstrated [43,44]. The difference between complement-mediated phagocytosis and FcγR mediated phagocytosis is discussed below in the phagosome formation section.

### 2.6. Dysregulation of Receptors during Phagocytosis in CF Macrophages

TLR5 is a crucial factor for the phagocytosis of *Pseudomonas aeruginosa* by macrophages [20]. Human CF MDMs have decreased TLR5 and subsequent impaired phagocytosis of *P. aeruginosa* [45]. However, in the same study, pharmacologic CFTR inhibition did not influence TLR5 expression in non-CF MDMs, suggesting that the loss of TLR5 in CF macrophages may be influenced indirectly by CFTR dysfunction, infection, or through structural changes not recapitulated by channel inhibition. No significant differences in the expression of TLR2 were observed in CF macrophages compared to control cells while an increased expression of TLR4 was observed in CF monocytes [46,47,48,49]. In addition to the aberrant localization at the PM, CF macrophages present altered intracellular TLR4 trafficking in response to activation. TLR4 is retained in the early endosome with reduced translocation to lysosomes resulting in delayed TLR4 degradation. These data demonstrate that CFTR in macrophages influences TLR4 spatial and temporal localization, thus determining the intensity and duration of the inflammatory response to LPS [49]. No change in the expression of TLR-4, TLR1/TLR-2, membrane CD14 (mCD14), CD16, and CD64 (FcγR) was observed in CFTR-deficient macrophages [45]. Yet, another study showed that mCD14 LPS receptors were reduced in CF macrophages and monocytes that impacted phagocytosis [50]. The reduced expression of mannose receptor CD206 and MARCO at mRNA and protein levels was observed in sputum macrophages of people with CF [51]. Decreased CD11b (CR3), an opsonic receptor, was also associated with reduced phagocytosis of *P. aeruginosa* [45,52]. Phagocytic receptors are also likely influenced by the CF airway milieu. A proteomics study [53] in the mouse macrophage cell line RAW 264.7 demonstrated numerous changes in phagocytic receptors and related proteins after exposure to neutrophil elastase, a prominent component in CF airways. Combined, these studies demonstrate the altered expression of receptors in CF macrophages associated with dysregulated phagocytosis, cellular activation, and increased susceptibility of people with CF to bacterial infections.

## 3. Phagosome Formation

### 3.1. Overview of FcγR-Mediated Normal Phagosome Formation

Phagocytosis occurs as a continuous and gradual process (Figure 1). Receptor activation and the activation of Syk (discussed previously) further activates actin polymerization and pseudopod formation, a downstream signaling pathway of phagocytosis. The actin cytoskeleton drives a conspicuous change in the shape of the phagocytic cell. The morphological changes assist in engulfing different particles of varying sizes greater than 0.5µm, wherein the particles are engulfed gradually into a plasma membrane-derived vesicle known as a phagosome [54].

The process of phagocytosis takes place in different steps; each step is morphologically distinguishable. Actin polymerization (formation of F-actin) is a critical step in phagocytosis. The first step is particle recognition by receptors and subsequent downstream signaling, during which predominant molecules are recruited and activated at the particle binding site that leads to F-actin formation [55]. F actin is assembled at the site of the particle; however, no clear protrusions are seen at this stage. The proteins that drive F actin formation are recruited to the particle binding site by an actin-independent ceramide and lipid raft enrichment that generate strained forces and bend the membrane inward [56]. The BARB domain-containing proteins such as formin-binding protein (FB17) and WASP-interacting protein (WIP) recognize the morphological change (membrane bend) and recruit WASP and ARP 2/3 proteins to the nascent phagocytic cup [55].

Once the proteins that drive F actin formation are assembled, new filament formation (F actin) begins with nucleation that depends on the interaction of nucleation-promoting factors (NPFs) with Arp 2/3, which is otherwise kept in an inactive state in the cell [57]. These NPFs belongs to the family of the Wiskott–Aldrich syndrome proteins (WASP). Different NFPs have been implicated in other functions in the vertebrate cell (lamellipodia formation, receptor-mediated endocytosis, endosome trafficking, etc.).

During phagocytosis, WASP-NPF and verprolin-homologous protein 2 (WAVE2-NPF) are the critical downstream effectors that control the dynamic organization of the actin cytoskeleton. The molecular structure of WASP consists of the verprolin homology domain–cofilin homology domain–acidic region (VCA domain), a polyproline domain, WASP homology 1 (WH1 domain), a primary area, and a GTPase-binding domain (GBD). The WASP domain exhibits an autoinhibited structure in which the VCA domain interacts with the GBD of the WASP itself. One of the mechanisms of WASP activation begins with WASP relaxation by the binding of the GTP-Cdc42/ SH3-containing protein (reviewed by Rougerie et al. [58]). Relaxation frees the VCA domain within the WASP protein. This allows the ARP 2/3 complex to bind to the VCA domain, leading to the addition of actin monomers and F-actin formation. Several WASP family proteins promote actin polymerization in an ARP 2/3-dependent (N -WASP, WAVE 1, 2, 3, and WASH) [59] and ARP 2/3-independent manner (JMY) [60].

Actin polymerization is also mediated by WAVE2 proteins present in mature macrophages [61]. The Abi1 complex interacts with the WH2 domain of WAVE2 and stabilizes WAVE2. Unlike WASP, the mechanism of WAVE2 activation is dependent on RAC and phospholipids, which frees the VCA domain of WAVE2 for ARP2/3 binding and nucleation to occur.

The second step in phagocytosis assumes a zipper model [62], wherein the plasma membrane is confined to the side of the particle to be ingested, generating a phagocytic cup. The drive to pseudopod formation (progression of F actin ring in a spatiotemporally controlled manner from the base to the tip of the phagocytic cup) and extension around the target particle is mediated by phospholipids [63]. Once the cup is formed, sealing followed by phagosome maturation occurs.

In addition to the WASP/WAVE2-mediated mechanism of actin nucleation and phagosome formation, the role of phospholipids is also critical for phagosome formation. Phosphoinositides (PIs) are phospholipids that are particularly essential during host–pathogen interactions. Several bacterial pathogens target PIs directly or indirectly to prevent phagosome formation, maturation, or to evade degradation [64,65,66]. The predominant PIs vary at each stage in the cup formation to generate F actin progression at the leading edge of the phagocytic cup [67,68,69,70]. The phospholipids activate/deactivate various proteins and enzymes required for phagosome formation, cup closure, and maturation at each stage. While actin nucleation and polymerization occur, actin depolymerization also occurs at the base.

While the above mechanisms concern phagocytic cup generation through FcγR-mediated receptor phagocytosis, there are both shared and unique elements between CR3- and FcγR-mediated phagocytosis. Cdc42 and WASP are not required for CR3-mediated phagocytosis, while Cdc42 is the principal component of FcγR-mediated phagocytosis [71,72]. RhoA plays a central role in CR3-mediated phagocytosis and activates formin-mediated unbranched F actin formation. The RhoA-formin axis also coordinates actin/microtubule dynamics in CR3-mediated phagocytosis. In contrast, in FcγR-mediated phagocytosis, ARP 2/3 mediates branched F actin formation [73,74].

### 3.2. Dysregulated Actin and Actin Regulating Proteins/Lipids during Infection with Opportunistic CF Pathogens

Overall, the process of actin remodeling and membrane protrusion in phagocytosis is very complex. Few studies have reported how actin dynamics are modulated for the benefit of bacteria in people with CF. This section focuses on the molecular dynamics of actin and actin-regulating proteins in macrophages that are altered during infections in CF. The bacteria discussed here are major opportunistic pathogens that cause severe pulmonary disease in CF.

*Staphylococcus aureus* is the major Gram-positive pathogen of CF lungs during infancy and in early childhood [75]. *S. aureus* protein A reduces the phagocytosis of *S. aureus* and *Pseudomonas aeruginosa* [76]. However, the role of *S. aureus* virulence factors in regulating actin dynamics during CF phagocytosis has not been explored.

*Burkholderia cenocepacia* is a Gram-negative bacterium that belongs to the *Burkholderia cepacia* complex. In CF, *B. cenocepacia* is highly transmissible from patient to patient and can cause a life-threatening sepsis-like picture known as ‘cepacia syndrome’, or result in hastened pulmonary deterioration [77]. Macrophages are the first line of defense against invading pathogens such as *B. cenocepacia* and are critical to initial and ongoing inflammatory responses. *B. cenocepacia* can survive intracellularly in macrophages, contributing to their persistence and virulence observed in CF [78,79]. The ability of *B. cenocepacia* to survive inside macrophages is attributed to disrupted actin cytoskeletal signaling [80,81,82]. Actin cytoskeleton depolymerization during *B. cenocepacia* infection changes the cell shape and results in a ball-on-a-string phenotype [80]. F actin appears as a cytoplasmic cluster within the macrophage in proximity to the intracellular bacteria within a membrane-bound phagosome [80,82]. Subsequently, infected macrophages appear rounded due to loss of focal contact associated with actin depolymerization. F actin formation during phagocytosis is coordinated by GTPases (RAC1 and Cdc42) of the Ras superfamily [83]. Macrophages infected with live *B. cenocepacia* show diminished RAC1 and Cdc42 activity and subsequently, reduced phagocytic activity [80]. One of *B. cenocepacia*’s key virulence factors is the type VI secretory system (T6SS), which inactivates RhoGTPases (RAC1 and Cdc42) and reduces the phagocytic capacity of macrophages [84,85].

Even though RhoGTPases are required for actin polymerization, Walpole et al. demonstrated that F actin cluster formation in the cytoplasm of macrophages is induced by RhoGTPase inhibition and is WASH dependent [82]. While the functional role of WASH involves endosomal trafficking and sorting, a shift in its function (cluster formation) in infected cells delays phagosome maturation. The authors demonstrated that macrophages infected with *B. cenocepacia* lack LAMP-1 in the late endosome/lysosome, indicating delayed phagosome maturation.

Although the manipulation of Rho GTPases and disturbance in actin dynamics caused by *B. cenocepacia* activate innate host immunity and protects mice from lethal infection, it does induce pyrin inflammasome activation. Aubert and colleagues demonstrated that a T6SS effector named tecA is a critical regulator of these processes [81]. The tecA protein induces irreversible deamidation of Asn-41 in macrophage Rho GTPases [81]. In bone marrow-derived macrophages, *B. cenocepacia*-induced RhoA inactivation is sensed by the pyrin inflammasome and stimulates caspase 1 activation, pyroptotic cell death, and IL-1β secretion [86]. While a tecA mutant strain failed to activate canonical inflammasome activation, re-expressing tecA induced pyrin inflammasome activation [81].

These data suggest that actin regulators act as a central switch to modulate phagocytosis by CF-relevant pathogens and other immune responses in macrophages. However, very few studies have focused on macrophage CFTR–actin interactions. One such study showing the interaction of actin (Ezrin) with CFTR is discussed below. Ezrin belongs to the ezrin-radixin-moesin proteins that bind F actin to CFTR through NHERF1 [87]. Ezrin also binds the SH2 domain of PI3K p85 required for PI3K/AKT signaling [88]. Di Pietro et al. showed that bone marrow-derived macrophages from CFTR deficient mice and MDMs from people with CF have less ezrin protein and poor localization in filopodia in response to LPS [89]. In addition, they demonstrated that decreased ezrin impaired AKT phosphorylation, with subsequent blunted PI3K/AKT activation and the negative regulation of TLR. Ezrin-deficient macrophages infected with *P. aeruginosa* had inhibited AKT signaling, increased pro-inflammatory cytokine production, and reduced anti-inflammatory responses. The response of ezrin-deficient macrophages infected with *P. aeruginosa* mimics the response of CF macrophages, highlighting the link between ezrin-CFTR, defective phagocytosis, and exaggerated inflammation. The reduced activation of PI3K/AKT signaling in CF macrophages may also weaken the non-opsonic phagocytic recognition of *P. aeruginosa* and response mechanisms during the early stage of CF lung disease [90]. Furthermore, alterations in calcium influx, which are necessary for membrane depolarization and the phagocytosis of *P. aeruginosa* in CF macrophages, are likely related to the PI3K-dependent recruitment of TRPV2 lipid rafts [91].

In addition to ezrin dynamics, PIs have also been linked to CFTR dysfunction. The intracellular distribution of PIs and high turnover/disappearance at each stage of the phagocytic cup mediates signaling events that aid in phagocytosis. PI5P localizes to the cellular membrane during phagosome formation to regulate actin remodeling and endosome vesicle trafficking. Delivery of liposomes loaded with PI5P in macrophages with defective CFTR improved phagocytosis, phagosome maturation, ROS production, and killing of *P. aeruginosa* [92]. In addition, PI5P delivery significantly reduced pro-inflammatory and increased anti-inflammatory cytokine production. The authors suggest that PI5P delivery activated macrophages with reduced neutrophil recruitment and associated pro-inflammatory damage.

Together, these studies indicate that defective macrophage phagocytosis is likely due to host-specific (actin and ezrin dynamics) mechanisms that can be compounded by specific pathogen-induced virulence factors (e.g., *B. cenocepacia*). However, most of the available data on macrophage actin dynamics were obtained using *B. cenocepacia*. Few studies have investigated other major pathogens or mixed infections, and information is lacking on how CFTR modulators impact these processes. Understanding how different CF opportunistic pathogens work cooperatively or antagonistically to take advantage of altered host cytoskeletal signaling may shed light on continued bacterial persistence in CF and offer new therapeutic strategies.

## 4. Normal Phagosome Maturation

After recognizing particles by receptors and cup formation, sealing of the cup takes place (phagosome). The maturation of the phagosome begins even before phagosome sealing occurs. The phagosome undergoes a cascade of fusion events with early endosomes, late endosomes, and lysosomes (Figure 2).

### 4.1. Early Phagosome: Fusion of Early Phagosome with Early Endosome

The early phagosome is mildly acidic (pH 6.5) with poor enzyme activity [93]. The first step after early phagosome formation is interaction with the early endosome. Rab GTPase acts as a molecular switch between an active (GTP-GEF bound state) and an inactive state (GDP-GDI bound state). Rabs are recruited and activated in the cytosol by the guanine exchange factor (GEF–displace GDP with GTP), while they are inactivated by the GTPase activating protein (GAP’s- hydrolyze GTP to GDI) present in the vesicular membranes. Once activated, they are moved to the vesicle membrane. The active RabGTP binds to effector proteins that regulate several steps in phagolysosome fusion or maturation [94].

The early endosome is loaded with several proteins such as Rab5, EEA1, SNARE, PI (3)P, and Rab 22a, along with GEFs such as RABEX-5 and Rabaptin-5 [13,95]. RABEX-5 and Rabaptin-5 (acts as a GEF) are acquired by the nascent phagosome and promote the activation of Rab5. Rabaptin-5 also recruits other molecules such as Vps34 (class III phosphoinositide 3-kinase vacuolar protein sorting 34) that enhance PI3P accumulation on the phagosome. PI3P and activated Rab5 recruit EEA1, promoting fusion of the early phagosome and endosome [95]. The activation of Rab5 by GEF also activates multi-subunit tethering complexes such as CORVET that brings the membrane together (tethering) and establishes the first step in the fusion. Mammalian CORVET (consisting of core subunits Vps 11, Vps 16, Vps 18, Vps 33, and two specific subunits, Vps3 and Vps 8) is only present on a subset of endosomes [96]. The soluble SNARE proteins are recruited by the Vps 33 subunit of the CORVET [97]. Fusion is mediated by the direct interaction of EEA1 with SNARE family proteins such as syntaxin 6 and syntaxin13 [95]. The EEA1 also binds NSF, rabex and rabaptin 5, and syntaxin13 [98,99].

Recycling: Although phagosomes undergo fusion events, the size of the phagosomes does not considerably change due to concurrent fission. Phagosomes recycle molecules to the plasma membrane through coat protein, Arf, and Rab GTPases [100]. The sorting nexin (SNX1, SNX2, SNX5, and SNX 6), vacuolar protein-sorting associated protein 26A (VPS26A), VPS 29, and VPS 35 retrieve cargo at the trans-Golgi network (TGN) [101]. Several Rabs (Rab4, Rab10, Rab11) mediate the transport of cargo proteins to the plasma membrane or TGN [102,103]. Finally, the endosomal sorting complex (ESCRT I and III) catalyzes the biogenesis of multivesicular bodies critical for cargo transport to the vacuole or lysosomes [104].

### 4.2. Late Phagosome: Late-Stage Phagosome Fusion with a Late Endosome

The late phagosome is characterized by a more acidic (5.5–6) environment. Upon subsequent maturation, the late endosome, like the phagosome, is associated with the loss of Rab 5 and acquisition of Rab 7. The mechanism associated with the transition to Rab 7 is not well understood. However, recent studies demonstrated that Mon1a/b and Ccz1b function as GEFs and regulate Rab 5 transition to Rab 7. Rab 7 acquisition is also mediated by the HOPS complex, while PI4P is associated with the recruitment of the HOPS complex [105]. Mammalian HOPS is well characterized by fusion assays and consists of seven proteins (Vps 11, Vps 16, Vps 18, Vps 33, Vps 39, and Vps 41) [96,106,107,108]. Fusion mediated by the HOPS complex takes place by replacing CORVET with the HOPS complex [109]. The late phagosome acquires LAMP 1 and LAMP 2 from the late endosome or Golgi complex that establishes fusion between the phagosome and the lysosome. The late phagosome also acquires enzymes through trans-Golgi transport vesicles [110]. Recycling and cargo transport for degradation also occur during the late phagosome stage and involves LBPA, ESCRT, and PDCD6IP [111].

### 4.3. Phagolysosome Formation: Fusion with the Lysosome

The late phagosome must be transported to the lysosome-rich perinuclear region to begin fusion with the lysosome. This process is executed by motor proteins such as dynein and dynactin. Activated Rab7 recruits downstream effector protein Rab interacting ribosomal protein (RILP) and oxysterol binding protein-related protein 1L (ORP1L). Next, the ORP1L–Rab7–RILP p150 Glued complex recruits the dynein motor [112]. Dynein translocates the phagosome towards the perinuclear cluster around the microtubule-organizing center (MTOC) [113].

Membrane fusion generally takes place in a cycle that begins with (i) tethering (involves HOPS) to bring late phagosomes and late lysosomes closer together to promote fusion, (ii) SNARE assembly, (iii) SNARE zippering, (iv) membrane fusion, (v) and SNARE disassembly and recycling [114]. There are 30 reported SNARE proteins to date; only a few specific SNARES form a stable trans-SNARE complex [96,114]. The SNARES for the membrane fusion of the late phagosome to the lysosome involves trans-SNARE Stx7-snap23-vam7/8 or Stx -Vtib-Stx8-Vamp 7/8 [114]. The trans SNARE complex establishes a conformational change, thereby forming a cis SNARE complex that brings the two membranes together in a zippering process [115]. HOPS docks proofread and prevent the disassembly of trans-SNARE [116,117,118]. αSNAP and NSF drive the disassembly of the SNARE complex through ATP hydrolysis, which frees the SNARE for reuse [119,120].

The phagolysosome is a highly acidic compartment (pH 4.5) facilitated by the vATPase. The vATPase is assembled in the late phagosomal membrane and transports a large amount of H+ into the lumen of the phagosomes. Most lysosomal hydrolases (proteases, nucleases, glycosidases, lipases, phospholipases, and phosphatase) require acidic pH for optimal enzyme activity [121]. The phagolysosome also incorporates the NADPH complex into the plasma membrane, generating reactive oxygen species such as superoxide anion, hydrogen peroxide, and hydroxyl radicals [122].

## 5. Specialized Vacuoles: Autophagy and LC3-Associated Phagocytosis (LAP) in CF

Autophagy is a self-conserved process that regulates inflammation activation and modulates NFκB activity and interferon production [123]. Autophagy plays a significant role in antigen presentation and the killing of extra and intracellular pathogens [124]. However, autophagy is also integral in recycling and clearing aggregated proteins and damaged organelles. There are multiple forms of autophagy, reviewed in depth by Klionsky et al. [125]. Macroautophagy (canonical autophagy) occurs inside a double membrane structure called an autophagosome. Multiple studies have demonstrated defective autophagy in CF phagocytes and other cells, with the critical restoration of macrophage properties through autophagy manipulation [126,127,128,129,130,131,132,133,134,135,136,137,138,139,140,141,142,143].

LC3-associated phagocytosis (LAP) is a recently described mechanism in which cells ingest particulate structures via non-canonical autophagy. LAP is differentiated from macroautophagy by the presence of a single-membrane vacuole and dependence upon RUN domain and cysteine-rich domain containing, Beclin-1-interacting protein (Rubicon). Upon stimulation by pattern recognition receptors, IgG receptors, or receptors recognizing dead cells, the components of the LAP pathway are recruited. Initiation does not require a pre-initiation ULK complex, making it distinct from macroautophagy. The class III PI3K complex composed of the core components Beclin-1, VPS34, UVRAG, and Rubicon induces the generation of PI3P in the membrane target of the LAPosome. Rubicon-containing PI3KC3 is essential for PI3P generation in the LAP pathway, while Rubicon has an inhibitory role in the canonical autophagy pathway. PI3P mediates the recruitment of ATG5-12 and LC3 PE ubiquitin-like conjugation systems. ATG4 processes LC3 to reveal a C terminal glycine referred to as LC3-I. ATG7 (acts like E1 activating) and ATG10 (acts like E2 conjugating) irreversibly conjugate ATG5 and ATG12. The ATG5–ATG12 conjugate binds ATG16L, and the multimeric complex ATG5-ATG12-ATG16L functions as an E3 ligase to conjugate PE to form LC3-II. The C terminal glycine is conjugated to the phosphatidylethanolamine (PE), referred to as LC3-II. The lipidated LC3 binds the LAPosome membrane to facilitate fusion with the lysosome. Rubicon recruitment of PIP3 is required for NADPH oxidase activation and the generation of ROS. ROS and Rubicon are required to conjugate LC3 to LAPosomes and subsequent association with LAMP-positive lysosomes [144], unlike macroautophagy. Overall, LAP is molecularly and functionally distinct. However, common machinery is shared with conventional phagocytosis and macroautophagy, and the final fusion with the lysosome remains the same for all three pathways.

### Inhibition of Phagosome Maturation in CF and Impairment of Antimicrobial Properties by Pathogens

*B. cenocepacia* is a highly virulent member of the *B. cepacia* complex. *B. cenocepacia* can alter phagosome formation in both CF and non-CF macrophages. Upon recognition of *B. cenocepacia*, EEA1, rab5, and PI3P are recruited, indicating that bacteria-containing vacuoles progress to the early phagosomal stage [145]. However, phagocytosed *B. cenocepacia* fails to accumulate LAMP1, which is a late endosomal marker, thereby indicating an arrested maturation process [145]. *B. cenocepacia*-containing vacuoles remain in contact with the early endosome with a pH of 6.4 [145] due to the delayed recruitment of vATPase onto the membrane vacuole [85]. Additionally, a delay in NADPH oxidase complex assembly was observed during *B. cenocepacia* infection [85]. Rab7 recruited to the late endosome is not active and demonstrates an inability to fuse with lysosomes [146]. *B. cenocepacia* recruits LC3 for autophagy, inhibits other autophagy components, and remains nascent in the autophagosome [128,147]. Finally, *B. cenocepacia* escapes the vacuole as indicated by a vacuole disruption marker galectin-3 [147].

Autophagy stimulation through pharmacologic and non-pharmacologic measures improves bacterial clearance in CF macrophages [126,127,128,129,131,137]. However, the impact of *B. cenocepacia* or other pathogens on LAP in CF macrophages has not been explored. During maturation, the phagosomes acquire several antimicrobial properties, as discussed earlier. Despite these antimicrobial mechanisms, bacteria can survive inside macrophages, which is accentuated in macrophages from people with defective CFTR. A recent study demonstrated that CFTR localizes with LC3-labeled autophagosomes at baseline in normal macrophages and autophagosomes and autophagolysosomes during infection [134]. In CF macrophages, the colocalization of *B. cenocepacia* with CFTR was significantly reduced. Further, defective CFTR was associated with failed lysosomal acidification. Interestingly, these results were specific to *B. cenocepacia*, as CFTR was not found to acidify *E. coli*-containing vacuoles. Prior work suggested both CFTR-dependent and independent roles in phagosome acidification [148,149,150]. Defective activation of autophagy [137] and, potentially, phagolysosome acidification [148,151] in CF macrophages also contributes to the impaired killing of *P. aeruginosa*. Further studies are needed to determine CFTR’s role in phagosome maturation and acidification during infection with other CF pathogens.

In addition to acidification defects, CF macrophage phagosomes also have alterations in NADPH oxidase assembly and subsequent ROS production. CF macrophages were found to have decreased protein kinase C-mediated phosphorylation of cytosolic NADPH oxidase components necessary for complex formation (p47*^phox^*, p40*^phox^*). These defects were independent of pathogens but were amplified by bacteria such as *B. cenocepacia* [152]. Similar findings were shown in CF neutrophils [153]. The altered formation of ceramide-enriched platforms at the plasma membrane of CF macrophages may also contribute to suboptimal clustering and the activation of the p47*^phox^* subunit [154], thus weakening ROS production and *P. aeruginosa* killing [155]. The non-tuberculosis *Mycobacterium abscessus* is an emerging opportunistic pathogen among people with CF [156]. Zebrafish studies suggest that CFTR-deficient macrophages show an increased burden of *M. abscessus* due to altered NOX2-dependent ROS production [157]. Combined, these studies indicate that CFTR deficiency leads to the increased growth of intracellular bacteria within macrophage phagosomes due to a poor activation of CFTR-dependent NOX2-mediated ROS production.

In murine macrophages, the rough strain of *M. abscessus* actively replicates within the phagosome and appears as a cluster of a few large phagosomes. Increased bacterial burden beyond phagosomal capacity leads to phagosomal rupture and the release of bacterial DNA into the cytosol. The active replication of mycobacterium and phagosomal rupture also causes mitochondrial damage and mtDNA release into the cytosol [158]. Mitochondrial oxidative stress enhances the production of Type I IFN and leads to increased virulence via cell to cell spreading [159]. The virulence of *M. abscessus* is also attributed to its ESX conserved component EccE4, which inhibits phagosomal acidification [160]. These data have not been confirmed in CF macrophage models, but ongoing work demonstrates that CF macrophages are more susceptible to the rough morphotype [161]. Continued insights into CFTR-dependent and pathogen-specific virulence mechanisms that subvert phagosome formation and activity may clarify why people with CF are more vulnerable to opportunistic pathogens such as NTM and *Burkholderia* species. A summary of CF macrophage phagocytic defects is shown in Figure 3.

Given the role of CFTR in immune modulation [162], a greater understanding of CFTR and its interaction with phagosomal formation proteins could provide mechanistic insight into why people with CF are susceptible to chronic respiratory infections. Studies investigating the interaction of CFTR and proteins such as SNARE and Rab are focused on CFTR trafficking in epithelial cells. Very few studies provide in-depth explorations of phagosomal protein-CFTR interchanges during CF host-pathogen interactions. However, the Rab7 protein, which is required for the proper aggregation and fusion of late endocytic structures [163], failed to increase during LPS stimulation in CF macrophages, suggesting an aberrant maturation of vesicles in the endosomal–lysosomal axis [49]. Formation of the CFTR and PTEN complex is also required for efficient intracellular *P. aeruginosa* killing in human monocytes [164], presumably via the PTEN-promoting dephosphorylation of Rab7 and regulating its localization to the late endosomal membranes [165]. Finally, one report shows how CFTR–SNARE interactions are critical to potentiating autophagy. Syntax 17 (Stx17) is required for fusion with the lysosome; lack of Stx17 causes the accumulation of autophagosomes without degradation [166]. In a study conducted by Arora et al. using epithelial cells, Stx17 was associated with CFTR at a late stage of autophagy to mediate bacterial clearance [133]. Whether these interactions are conserved in CF macrophages has not yet been determined. A few reports are available on the role of Rabs in other CFTR-defective cells. For example, altered ion homeostasis (increased sodium and chloride, decreased magnesium) leads to the inactivation of Rab27a and impaired degranulation in CF neutrophils [167]. In human bronchial epithelial cells, CFTR depletion decreases the availability of Rab5, EEA1, and PI3P which are required for endosomal fusion maturation and trafficking [168]. More work is needed to connect these findings across different cell types, including macrophages, and link all the varied phagocytic defects present in CF.

## 6. CFTR Modulators/Other Therapeutics and CF Phagocytosis

Drugs commonly used to treat CF include anti-inflammatory agents, pancreatic enzymes, mucolytics, and antibiotics [169,170]. With the discovery of CFTR modulators that partially restore CFTR expression and function, a paradigm shift has occurred in the treatment of CF. CFTR modulators have various efficacies, targeting CFTR based on CFTR variant classes. Drugs currently approved for different variants include ivacaftor (Kalydeco^®^), lumacaftor/ivacaftor (Orkambi^®^), tezacaftor/ivacaftor (Symkedo ^®^ or Symkevi^®^) and elexacaftor/tezacaftor/ ivacaftor (Trikafta^®^ or Kaftrio^®^). The effects of these drugs on only CF monocytes and macrophages are discussed below.

### 6.1. CFTR Modulators

Iron regulation plays a critical role during macrophage-pathogen interactions. The transferrin receptor 1 (TFR1) transports iron from the extracellular milieu. Iron favors the growth of bacteria, particularly biofilm formation by *P. aeruginosa,* a common CF pathogen [171]. CFTR modulators can influence the expression of iron regulatory proteins. Lumacaftor/ivacaftor treatment reduced CF macrophage TFR1, transferrin and improved lactoferrin expression [172]. A high content of total iron was observed in conditioned media from CF macrophages. However, lumacaftor/ ivacaftor treatment reduced total iron in the conditioned media [172]. Whether changes in CFTR modulator-mediated iron transport alters phagolysosomal iron transporters, macrophage polarization induced by intracellular iron equilibrium, or other aspects of iron-mediated immunometabolism necessary for proper phagocytosis and killing of pathogens remains unknown.

A few studies examined the direct CFTR modulator impacts upon macrophage phagocytosis. Lumacaftor alone increased the ability of CF macrophages to kill *P. aeruginosa*. However, its combination with ivacaftor reduced the potential of lumacaftor to phagocytose and kill *P. aeruginosa.* In contrast, cytokine secretion was unaltered in CF macrophages treated with lumacaftor while ivacaftor or lumacaftor/ivacaftor reduced pro-inflammatory cytokines in CF MDMs [173]. This study suggests potential negative interactions of CFTR modulator components on phagocytosis. A separate study corroborated these findings and found that MDMs from people with CF and ivacaftor-sensitive CFTR variants had improved phagocytosis, reduced apoptosis, and decreased pro-inflammatory cytokines after ivacaftor treatment. However, MDMs from people with F508del variants treated with lumacaftor/ivacaftor showed decreased phagocytosis in non-CF and CF macrophages and had low M1 polarized monocytes [174]. Ongoing work from our labs and others will hopefully shed light on the impact of newer generation CFTR modulators upon macrophage phagocytosis.

In addition to potential benefits of CFTR modulator on bacterial phagocytosis, a recent study demonstrated that CFTR regulates the acidification of autophagolysosomes and their subsequent proteolytic degradative functions. Treatment with tezacaftor/ivacaftor improved lysosomal acidification, autophagy flux, and bacterial clearance in CF murine macrophages [134]. Targeting specific lysosomal activities in CF remains a therapeutic target of interest.

Several studies examined phagocyte surface receptor changes in the context of CFTR modulation. Ivacaftor treatment reduced the cell surface activation marker CD63 on monocytes obtained from people with CF who carried the G551D variant. Treatment with ivacaftor resulted in monocytes with reduced response to PMA based on CD11b and CXCR2 expression compared to pre-treatment [175]. Ivacaftor was also shown to induce changes in the plasma membrane proteome of G551D monocytes [176]. Ivacaftor increased CF monocyte migration proteins such as ENO1 and PFN1 and leukocyte migration proteins such as ICAM3 and CORO1A. Other proteins involved in inflammation such as S100A9, MX1, and HLA-B were reduced in monocytes with ex vivo ivacaftor treatment. A limited description of specific affected proteins involved in phagocytosis was provided, however proteins such as LEMD2 (transmembrane adapter for ESCRT) and ACTG1 were altered by ivacaftor, suggesting a potential impact on degradative and actin-related processes. In a separate proteomics study, several proteins involved in the regulation of actin cytoskeleton and leukocyte trans-endothelial migration were strongly downregulated in CF monocytes carrying residual function CFTR variants after treatment with ivacaftor [177]. In a fourth study, people with CF treated with lumacaftor/ivacaftor had a lower expression of CXCR2 (receptor for IL-8) on CD14+ CD16- monocytes compared to healthy controls. Discontinuation of treatment was associated with increased CXCR2 surface levels in CD14+ CD16- CF monocytes [178]. Transcriptomic studies of CF monocytes with the R117H variant after 7 days of ivacaftor treatment showed an upregulation of inflammatory pathways including antigen processing, cell cycle oxidative phosphorylation, and the unfolded protein response [179]. Interestingly, the Rho guanine nucleotide exchange factor TIAM1 was decreased in expression after treatment. The significance of this change in relation to cytoskeletal activities in CF monocytes and MDMs post modulator treatment remains undetermined. A transcriptomics study of whole blood RNA-Seq profiles after 3 months of treatment with lumacaftor/ivacaftor demonstrated changes in several pathways related to phagocytosis, including RhoA signaling and actin signaling. These pathway changes were independent of the clinical response to therapy [180]. However, none of the studies described looked at the influence of CFTR modulators upon alveolar macrophage surface receptors or phagocytic signaling pathways. Other studies examined the influence of CFTR modulators on inflammatory signaling in monocytes and macrophages and were reviewed elsewhere [181,182].

### 6.2. Alternative Agents: Epigallocatechin-3-Gallate, Cysteamine, and Roscovitine

Many alternative agents that restore CFTR expression or impact pathways downstream of CFTR are under investigation. The natural compound epigallocatechin-3-gallate (EGCG) is a polyphenol found in green tea whose extract has anti-inflammatory properties. EGCG was studied in CF as a demethylation agent that restores the expression of ATG12 (autophagy regulator) by inhibiting the methylation of its promoter. CF macrophages demonstrated reduced expression of an ATG5-ATG12 protein complex with significant methylation of ATG12. Treatment with EGCG improved mice macrophage CFTR function, reduced ATG12 promoter methylation, improved autophagy activation, and reduced *B. cenocepacia* burden [183].

Cysteamine is a water-soluble FDA-approved drug that is used to treat nephropathic cystinosis [60] and is under investigation as a CF therapeutic. Cysteamine’s impacts include antioxidant [184], anti-inflammatory [137], autophagy induction [185,186], anti-bacterial [131], mucolytic, and anti-biofilm [187,188] properties as reported in several cell types. Here, we discussed only the effects of cysteamine on CF macrophages. Cysteamine, a transglutaminase 2 (TG2) inhibitor, prevents cross-linking of CFTR and essential autophagy proteins into aggresomes and restores autophagy. Bone marrow-derived macrophages from CF mice pretreated with cysteamine increased the internalization of *P. aeruginosa* through BECN1 expression, a critical component of autophagy induction. In addition, cysteamine also reduced inflammatory cytokines during *P. aeruginosa* infection [137]. A similar effect was observed in MDMs derived from people with CF when treated with cysteamine. Cysteamine decreased TG2 expression, improved LC3II expression (marker of autophagosomes) and decreased beclin-1 and P62 aggregation. Cysteamine increased the colocalization of *B. cenocepacia* with LC3-labeled autophagosomes and reduced the bacterial load. The direct antibacterial effect of cysteamine against various CF pathogens was further reported [131]. Cysteamine has also been used in studies with modified delivery strategies. The terminal amine group of a polyamidoamine dendrimer modified with a cysteamine-like structure with a sulphydryl group termed PAMAM-DEN^cys^ improved autophagy and rescued CFTR from aggresomes resulting in reduced *P. aeruginosa* infection in CF cells. PAMAM-DEN^cys^ was also reported to have bactericidal and mucolytic properties [187].

Roscovitine and its metabolite M3 belong to a family of cyclin-dependent kinase (CDK) inhibitors. Roscovitine previously underwent safety and efficacy studies in CF. In ex vivo studies, roscovitine significantly reduced the bacterial load of several multi-drug resistant pathogens in CF MDMs, like cysteamine’s effects [189]. Cysteamine combined with increasing doses of roscovitine also showed a dose-dependent reduction in bacterial load. Interestingly, roscovitine did not directly affect phagocytosis, but was partially dependent on CFTR stabilization and activation of the Ca^2+^ channel TRPC6. In addition, roscovitine, combined with tezacaftor and ivacaftor, significantly reduced bacterial load compared to their individual effects [189].

Several other therapeutic studies are ongoing in relation to CF macrophages, including the use of microRNAs to regulate critical signaling pathways [126,130]. Combined, these studies suggest that alterative therapeutics agents may have beneficial effects on targeted pathways of CF macrophage phagocytosis.

## 7. Conclusions

Macrophages are the first line of defense against invading pathogens, the ensuing inflammatory response and in the subsequent resolution of inflammation. Phagocytosis is a crucial mechanism by which macrophages contain infections and inflammation. Numerous studies demonstrate the defective clearance of pathogens by CF macrophages. These defects are associated with different stages of phagocytosis, from internalization to maturation. A unifying mechanism behind defective phagocytosis in CF is still unclear. Few studies focused on interactions between CFTR and proteins involved in different stages of phagocytosis, and little is known about comparative responses between different types of resident and recruited macrophages. Most macrophage dysfunction studies focused on single pathogens such as *B. cenocepacia* or *P. aeruginosa*, with few studies focused on other CF pathogens or mixed-pathogen models. Macrophages are now considered a potential therapeutic target in CF, with several studies showing the efficacy of CFTR modulators. Therefore, understanding alterations in phagocytosis induced by CFTR dysfunction might assist in developing novel therapeutics for all people with CF.

## Figures and Tables

**Figure 1 ijms-23-07750-f001:**
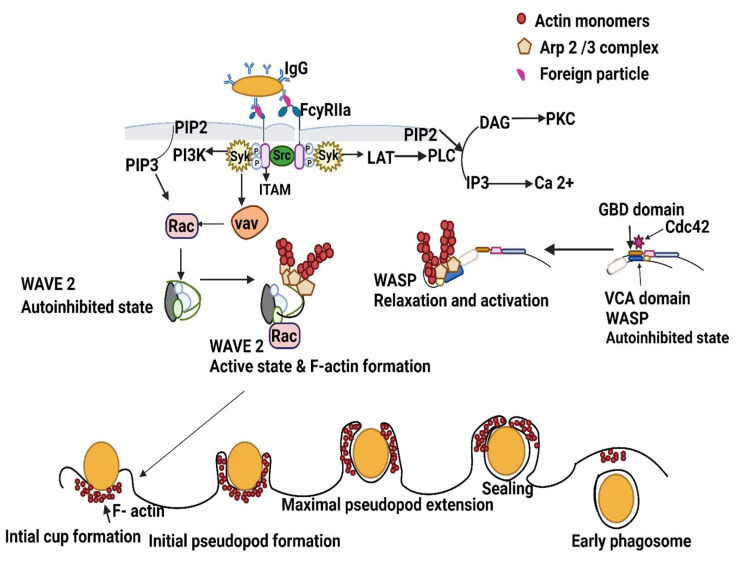
Schematic representation of phagosome formation. FcγRIIa crosslinks with IgG bound to foreign particles and induces the activation of Src family kinases, which phosphorylate the tyrosine residues in ITAMs. Then, Syk and phosphorylated ITAMs phosphorylate and activate LAT (linker for T cells activation). LAT, in association with phospholipase C gamma (PLCγ), produces inositol trisphosphate (IP3) and diacylglycerol (DAG), causing calcium release and the activation of protein kinase C (PKC), respectively. Syk recruits and activates the phosphatidylinositol 3-kinase (PI3K) phosphatidylinositol-3,4,5-trisphosphate (PIP3) at the phagocytic cup. Syk also activates the guanine nucleotide exchange factor Vav, which activates the GTPase Rac. Further, activated Rac and CDC42 activate WAVE and WASP, respectively, from an autoinhibited state to an active state. Activated WAVE and WASP promote actin monomer and Arp 2/3 complex binding to the VCA domain that leads to F-actin formation. Assembly of F-actin nucleation components and F actin formation proceeds with initial cup formation, pseudopod formation, maximal pseudopod formation, sealing or closure, and results in early phagosome formation.

**Figure 2 ijms-23-07750-f002:**
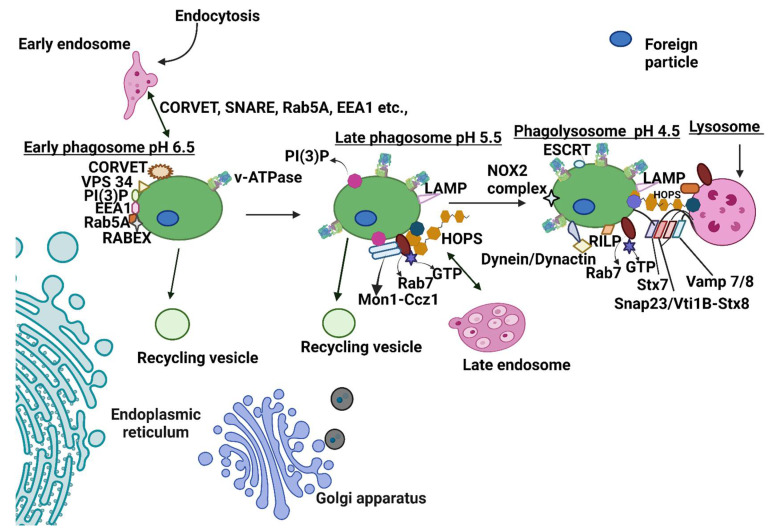
Schematic representation of phagosome maturation in parallel with endosome maturation. Phagosome maturation involves continuous fusion and fission events with early endosome, late endosome, lysosome, and recycling vesicles that cause the acquisition and loss of different markers of maturation. Following early phagosome formation, sequential steps form late phagosomes and phagolysosomes. The early phagosome is marked by the presence of Rab5, EEA1, VPS34, and PI(3)P. PI(3)P and GTP-bound Rab5 promote the recruitment of EEA1 that facilitates fusion between early phagosomes and early endosomes. Activated Rab5 also recruits the CORVET complex to promote fusion with the early endosome. The late phagosome is marked by the presence of Rab7 and LAMPs. In the late phagosome, PI3P recruits and activates the GEF, Mon1-Ccz1 complex, and activates Rab7. HOPS also activates Rab7 in the late phagosome. Rab7 is essential for phagolysosome fusion. The trans-SNARE complex also mediates the fusion between the late phagosome and lysosome. HOPS guides the late phagosome to the lysosome and stabilizes the trans-SNARE Stx7-Snap23-Vamp7/8 or Stx7-Vti1b-Stx8-Vamp7/8. The phagolysosome is rich in many degradative enzymes such as proteases, cathepsins, lysozymes, and NADPH oxidase complex. V-ATPase also accumulates and makes the lumen more acidic as the phagosome matures.

**Figure 3 ijms-23-07750-f003:**
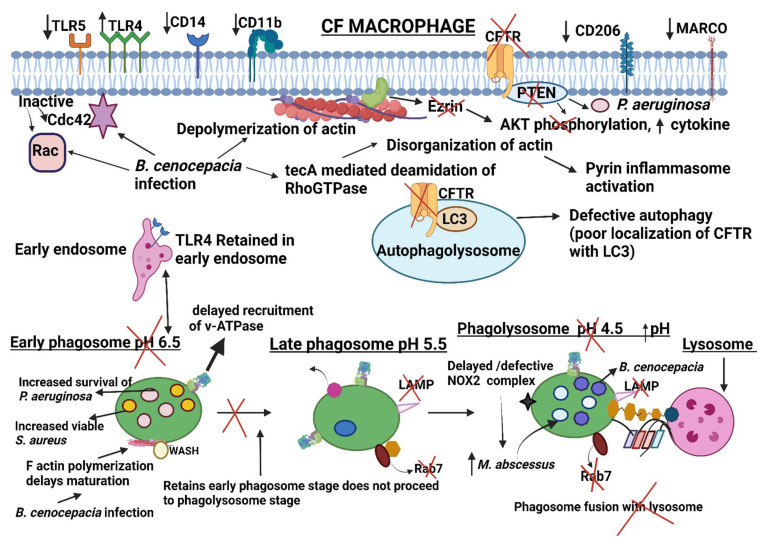
Schematic representation of phagocytosis defects in CF macrophages. Dysregulation of surface receptors, actin cytoskeleton dynamics, phagosome maturation, and autophagosome formation are shown during infection in CF macrophages. (1) Decreased expression of receptors such as TLR5, CD14, CD11b, CD206, and MARCO are displayed in CF macrophages. (2) Dysregulation of the actin cytoskeleton due to inactive Cdc42, Rac, and tecA mediated disorganization of actin during *B. cenocepacia* infection is shown. Ezrin links the actin cytoskeleton to the plasma membrane. (3) Poor ezrin expression in CF macrophages leads to decreased AKT phosphorylation and increased cytokine expression. Increased TLR4 expression and retained TLR4 in early endosomes is demonstrated. (4) Dysregulation of phagosome maturation is shown in CF macrophages. Delayed recruitment of V-ATPase, increased acidification of phagosomes, and F actin polymerization of phagosomes prevents maturation of phagosomes. During *B. cenocepacia* infection, the phagosome proceeds to the early phagosome stage but does not mature to the phagolysosome stage and bacteria survive inside the phagolysosome and prevent phago-lysosomal fusion. (5) Defective colocalization of CFTR with LC3 is observed in autophagosomes leading to defective autophagy. Increased survival of *P. aeruginosa*, *S. aureus*, and *B. cenocepacia* in CF macrophage result from defective autophagy. (6) Absence of the CFTR-PTEN complex decreases Akt phosphorylation and thereby increases pro-inflammatory cytokines. Also, insufficient amounts of active PTEN contribute to defective clearance of intracellular *P. aeruginosa*.

## Data Availability

Not applicable.

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
