# Peer review of "Emerging Concepts in Defective Macrophage Phagocytosis in Cystic Fibrosis"

_ijms, 2022, doi:10.3390/ijms23147750_

Round 1
Reviewer 1 Report
"Emerging Concepts in Defective Macrophage Phagocytosis in Cystic Fibrosis" is a review manuscript for consideration in International Journal of Molecular Sciences by Jaganathan, Bruscia, and Kopp.
This is a comprehensive review of how macrophage function can be impaired in CF, either as a direct consequence of impaired CFTR function/trafficking or as an indirect consequence of bacterial infection.
The review is well-referenced and up to date. I have only a few minor points.
1. The figures contain a few objects, for instance Rac, that are difficult to read because they are black font on blue background. This will probably not print well on grayscale and would be difficult to see if projected on a screen. I recommend modifying the figures to make objects more legible. In figure 1, it looks like the Fc portion of the antibody is bound to the foreign particle, rather than Fab.
2. Some of the font sizes in the figures are small and could be difficult to read because they are crossed out or overlap with other images.
3. The discussion of how pathogens interfere with phagocytosis focuses mostly on Gram-negative pathogens. It may be important to acknowledge Staphylococcus aureus has Protein A. This binds the Fc portion of antibodies, inhibiting immune recognition and phagocytosis.
4. Line 84: "spingosine one phosphate" - should this be "sphingosine-1-phosphate"
5. Lines 112-117: "loss of TLR5 in CF macrophages may be influenced indirectly by CFTR dysfunction or through structural changes..." One of the indirect causes could be infection.
6. The use of marketing names for CFTR modulator drugs - Is this necessary or required by the journal?
Author Response
- The figures contain a few objects, for instance Rac, that are difficult to read because they are black font on blue background. This will probably not print well on grayscale and would be difficult to see if projected on a screen. I recommend modifying the figures to make objects more legible. In figure 1, it looks like the Fc portion of the antibody is bound to the foreign particle, rather than Fab.
R: Thank you for your time and effort. We have we revised the figures according to suggestions. The new image shows the antibody (Fab portion) and receptor interaction.
- Some of the font sizes in the figures are small and could be difficult to read because they are crossed out or overlap with other images.
R: Revised, as above to improve font sizes for defective pathways in Fig 3.
- The discussion of how pathogens interfere with phagocytosis focuses mostly on Gram-negative pathogens. It may be important to acknowledge Staphylococcus aureus has Protein A. This binds the Fc portion of antibodies, inhibiting immune recognition and phagocytosis.
R: Agree, there is a lack of CF literature regarding S. aureus impact upon CF phagocytosis, but we have added the potential impact above to the beginning of the discussion on CF pathogens.
- Line 84: "sphingosine one phosphate" - should this be "sphingosine-1-phosphate"
R: corrected
- Lines 112-117: "loss of TLR5 in CF macrophages may be influenced indirectly by CFTR dysfunction or through structural changes..." One of the indirect causes could be infection.
R: Infection added to the sentence.
- The use of marketing names for CFTR modulator drugs - Is this necessary or required by the journal?
R: We left this so readers can easily identify the drugs if they are more familiar with the trade names.
Reviewer 2 Report
In this work, Jaganahan and colleagues made an excellent literature review on the function and role in CF of the macrophages. They described in deep details the phagocytosis mechanism, its dysregulation in CF ariway epithelial cells, and the effects of CFTR modulators. This work significantly contributes to the knowledge on CF inflammation and its regulation.
Author Response
In this work, Jaganathan and colleagues made an excellent literature review on the function and role in CF of the macrophages. They described in deep details the phagocytosis mechanism, its dysregulation in CF airway epithelial cells, and the effects of CFTR modulators. This work significantly contributes to the knowledge on CF inflammation and its regulation.
R: Thank you for your time and effort.